# Mapping of Allergenic Tree Species in Highly Urbanized Area Using PlanetScope Imagery—A Case Study of Zagreb, Croatia

Mateo Gašparović [1], Dino Dobrinić [2,*] and Ivan Pilaš [3]

1   Chair of Photogrammetry and Remote Sensing, Faculty of Geodesy, University of Zagreb, 10000 Zagreb, Croatia; mgasparovic@geof.unizg.hr
2   Chair of Geoinformatics, Faculty of Geodesy, University of Zagreb, 10000 Zagreb, Croatia
3   Croatian Forest Research Institute, Division of Ecology, Cvjetno naselje 41, 10450 Jastrebarsko, Croatia; ivanp@sumins.hr
*   Correspondence: ddobrinic@geof.unizg.hr; Tel.: +385-1-4639-369

**Abstract:** Mapping and identifying allergenic tree species in densely urbanized regions is vital for understanding their distribution and prevalence. However, accurately detecting individual allergenic tree species in urban green spaces remains challenging due to their smaller site and patchiness. To overcome these issues, PlanetScope (PS) satellite imagery offers significant benefits compared with moderate or high-resolution RS imagery due to its daily temporal resolution and 3 m spatial resolution. Therefore, the primary objectives of this research were to: assess the feasibility of mapping allergenic tree species in the highly urbanized area using high-resolution PS imagery; evaluate and compare the performance of the most important machine learning and feature selection methods for accurate detection of individual allergenic tree species. The research incorporated three classification scenarios based on ground truth data: The first scenario (CS1) used single-date PS imagery with vegetation indices (VI), while the second and third scenarios (CS2 and CS3) used multitemporal PS imagery with VI, and GLCM and VI, respectively. The study demonstrated the feasibility of using multitemporal eight-band PlanetScope imagery to detect allergenic tree species, with the XGB method outperforming others with an overall accuracy of 73.13% in CS3. However, the classification accuracy varied between the scenarios and species, revealing limitations including the inherent heterogeneity of urban green spaces. Future research should integrate high-resolution satellite imagery with aerial photography or LiDAR data along with deep learning methods. This approach has the potential to classify dominant tree species in highly complex urban environments with increased accuracy, which is essential for urban planning and public health.

**Keywords:** urban green area; random forest; neural network; extreme gradient boosting; PlanetScope; feature selection

## 1. Introduction

Remote sensing utilizes satellite and/or airborne-based sensors that can provide useful data for inventorying vegetation [1]. The population of individuals who have allergies to pollen has been consistently increasing, particularly in urban and industrial areas [2]. Allergies are the most prevalent chronic disease in Europe, affecting over 150 million European citizens, according to the European Academy of Allergy and Clinical Immunology (EAACI). It is projected that by 2025, more than 50% of all Europeans will suffer from at least one type of allergy, without distinction of age, social class, or geographical location. The prevalence of allergic diseases is increasing rapidly in parallel with triggers such as urbanization, industrialization, pollution, and climate change [3]. These diseases lead to a reduction in productivity and an increase in sick leave among sensitive individuals, greatly impacting European business and healthcare economies. It is estimated that asthma and allergic rhinitis alone result in over 100 million lost workdays and missed school days in Europe annually [4].

Identifying and locating allergenic tree species in highly urbanized areas is crucial as it allows for an understanding of their distribution and prevalence in urban areas. Urbanization can lead to changes in the tree species composition, which can significantly impact the health of residents, particularly those with allergies [5]. Pollen is defined as the strongest natural aeroallergen and the most common causative agent of allergic diseases of the respiratory system in Europe. The geographical distribution of plants has a significant influence on their allergenic impact, e.g., the trees of the family Betulaceae are widespread and shed large quantities of windborne pollen [6]. Peternel et al. [6] made an analysis of the pollen count distribution for the central part of Croatia (Zagreb). As mentioned in their research, the most common allergenic species in Croatia are particular tree species such as alder (*Alnus* sp.), hazel (*Corylus avellana* L.), and birch (*Betula pendula* L.), weeds such as *Ambrosia* sp., *Parietaria* sp., and *Artemisia* sp., and many types of grass, e.g., *Poaceae*. In addition, these allergenic species are common in the temperate zones of Europe [7], and the developed methodology from this research could also be applicable to other areas of Europe. Furthermore, by identifying and mapping allergenic tree species, city planners and public health officials can make informed decisions about tree planting and management [8].

Many studies used optical remote sensing satellite data to map allergenic pollen vegetation on a large scale. Manzini et al. [9] developed the FlorTree model to evaluate the pollution-removal capabilities of 221 urban tree species in various cities. The model considers species-specific and local factors to maximize air quality improvements. Their findings emphasize the necessity for strategic urban tree selection based on local conditions. From the UK pollen network, McInnes et al. [10] used 12 major species and produced detailed maps of pollen-producing plants' locations. The results obtained demonstrate the varying geographical patterns of allergies and allergic asthma, and the maps can be utilized to investigate environmental exposure and its impact on human health. Integration of remote sensing data (i.e., Rapid Eye, Landsat-8, and Sentinel-2 imagery) with in situ pollen measurements was investigated by Lugonja et al. [1]. In this research, a top-down approach that combines the distribution of suitable habitats and airborne pollen concentrations was used, detailed crop classification maps were produced, and it was found that the strongest correlation between pollen and variation was in areas with high soya bean and sugar beet cultivation. Furthermore, Lara et al. [11] examined trends in annual *Platanus* pollen concentrations in central Spain from 2003 to 2019. The findings suggest that variations in the diversity and abundance of allergenic tree species in urban green spaces may be responsible for the trends observed in the dynamics and behavior of airborne pollen from these species. However, using remote sensing data to accurately identify individual allergenic tree species is difficult due to their small area and patchiness in urban green spaces [12]. To address this challenge, a key objective of this research will be to utilize high-resolution satellite imagery for mapping allergenic tree species in urban areas, since the mapping of individual trees becomes more feasible from a resolution of 3 m or higher [13].

In the past, mapping urban green areas usually employed field measurements [7], which are time-consuming, labor-intensive, and ineffective since these areas change rapidly over time. Currently, remote sensing (RS) technology has proven to be an efficient method for the classification of urban green spaces [14]. Depending on the spatial resolution of the RS dana used and the extent of the study area investigated, low-resolution imagery (MODIS; [15,16]), with moderate resolution (Landsat, Sentinel-2; [17–19]), and with the advancement of the satellite sensors with very high-resolution (WorldView; [20,21]), has been utilized to acquire rapid and accurate information over green areas. However, the temporal frequency of the above-mentioned satellite imagery may not be sufficient to capture the variations in green spaces. In addition, the spatial resolution may not be detailed enough to capture variations in urban green volumes accurately [18]. To overcome these issues, PlanetScope (PS) satellite imagery [22] offers significant benefits over previous RS data due to its daily temporal resolution and 3 m spatial resolution, providing an effective solution to overcome existing issues. Hence, PS data has been successfully used, for example, in mapping rubber plantations [23], snow-covered areas in forested mountain

ecosystems [24], mapping of lava flows in barren regions [25], and land-cover classification in urban areas [20]. Furthermore, in 2019, PlanetScope announced the general commercial availability of the next generation of PS Monitoring product, which includes eight spectral bands, instead of previously offering four bands [26]. Therefore, the performance of the eight-band PS imagery is investigated in this research for tree species in a highly urbanized area.

Image classification methods (e.g., unsupervised, supervised) are usually used to map urban green areas. Mostly used unsupervised classification methods include ISODATA, or k-means clustering [21,27,28], whereas the most important supervised classification methods for urban green area mapping are random forest (RF), support vector machine (SVM), and neural networks (NN). Puissant et al. [29] used an RF classifier and an object-oriented approach for tree mapping in urban areas. The tests conducted using RF showed that this classifier is highly effective in identifying wooded vegetation in terms of user's and producer's accuracy. Furthermore, Chen et al. [30] proposed a method that utilizes neural networks and crowdsourced geospatial big data to automatically map urban green spaces (UGS). This method increased the user's accuracy in UGS mapping using Sentinel-2 imagery with 10 m spatial resolution. Deep learning (DL) methods such as convolutional neural networks (CNN) have been widely utilized for image classification in recent years. Xu et al. [14] presented a DL method for UGS mapping based on the phenological features using very high-resolution satellite imagery (i.e., GaoFen-2). The results indicate that the above-mentioned method can efficiently solve the misclassification problem of evergreen and deciduous trees. Furthermore, the integration of multiple sources and multi-spectral RS datasets leads to the creation of large-scale datasets for classification. However, these datasets often contain highly correlated features, which can introduce noise and negatively impact the classification performance [31]. As a result, different feature selection (FS) methods have been created that simplify the model and consequently help to accelerate the classification. Dobrinić et al. [32] compared various FS methods for vegetation mapping using multitemporal Sentinel-1 and Sentinel-2 imagery and additional data derived from the RS imagery.

As mentioned above, still most studies concentrate on allergenic pollen vegetation mapping on a large scale and the identification of individual allergenic tree species in urban green spaces is difficult due to their small size and patchy distribution. Hence, the primary goals of this study are to:

1. Assess the feasibility of mapping allergenic tree species in the highly urbanized area using high-resolution PlanetScope imagery;
2. Compare and evaluate the effectiveness of the most important ML and FS methods for the accurate detection of individual allergenic tree species.

The remainder of this paper is structured as follows: In Section 2, the research area and datasets utilized in this study are discussed. Section 3 explains the various classifiers and feature selection techniques used for identifying allergenic tree species. The results of the study and their analysis are presented in Section 4. Lastly, the key findings and conclusions are outlined in Section 5.

## 2. Materials

### 2.1. Study Area and Reference Data

In this research, the urban area of the city of Zagreb (Figure 1) was used for the accurate detection of individual allergenic tree species, with a focus on protected green areas, such as Botanical Garden, park Zrinjevac (Figure 1—Example subset 1), Lenuzzi's green "horseshoe", and parks with developed and cultivated green infrastructures, e.g., Newlyweds Park (Figure 1—Example subset 2). According to the last population census from 2021, the city of Zagreb has 769,944 inhabitants, with a positive yearly increase in the number of inhabitants. Therefore, an extent of 27.84 square km (5.80 × 4.80 km) was used in this research, mostly focusing on the central urban part of the city of Zagreb. Zagreb's

climate is classified as an oceanic climate (Cfb) [33] with warm summers and an average annual precipitation and temperature of 840 mm and 11.6 °C, respectively.

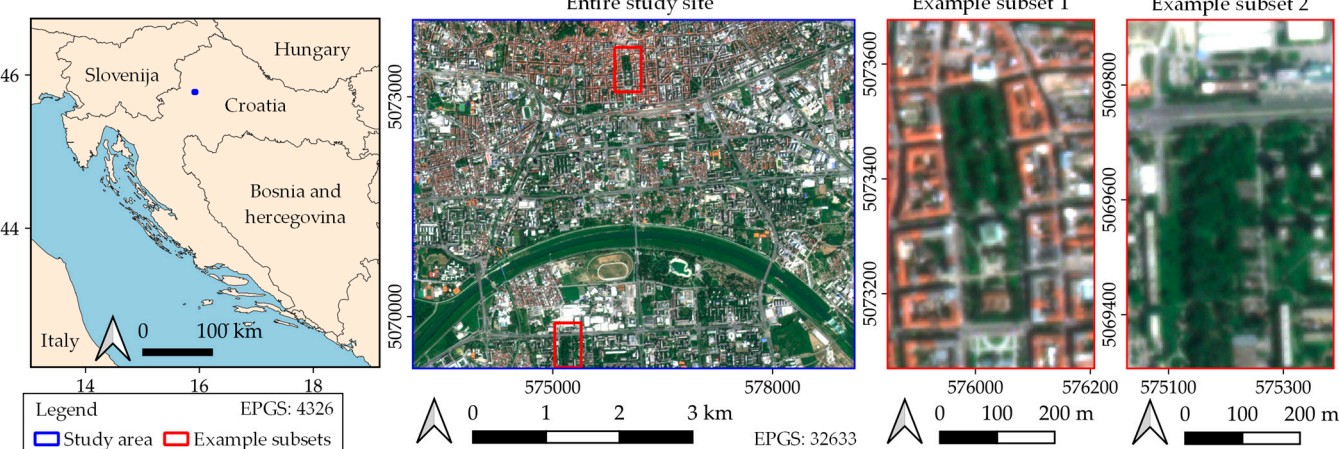

**Figure 1.** Location and geographic extent of the study area with a PS true-color composite image as the background. The red rectangles represent examples of subset 1 and 2 of urban green areas with protected green areas and cultivated green infrastructures, respectively.

The forest vegetation in the Zagreb county is characterized primarily by forests of pedunculate oak, narrow-leashed ash, black alder, willows, and poplars [34]. In the urban area of the city of Zagreb, birch, sycamore, plane tree, and pine are the most dominant species. According to Peternel et al. [2], the most allergenic plant groups are: alder, hazel, yew/cypress, birch, ash, hornbeam, grasses, elder, nettles, sweet chestnut, artemisia, and ambrosia.

For this research, the Web Map Service (WMS) of the Green Cadastre application was used as the reference dataset, which includes data on public green areas, playgrounds, and park equipment. The original dataset consists of 61,564 sample points, and after data filtering (e.g., trunk thickness higher than 10 cm, tree height higher than 5 m, etc.), 11,430 sample points were included as reference point data in a final model. According to Peternel et al. [2], three major allergenic tree species were chosen in this research, whilst the remaining tree species were categorized as other (Table 1).

**Table 1.** List of the allergenic tree species, along with genera/families of the species and a number of samples used for classification purposes.

| ID | Specie | Genera | Sample Count |
|----|--------|--------|--------------|
| 1 | Ash | *Fraxinus Americana, Fraxinus Excelsior Diversifolia, Fraxinus Ornus, Fraxinus Excelsior Globosa, Fraxinus Excelsior, Fraxinus Angustifolia* | 2709 |
| 2 | Birch | *Betula Pubescens, Betula Pendula, Betula Papyrifera* | 2808 |
| 3 | Plane tree | *Platanus × Acerifolia* | 2693 |
| 4 | Other | *Alnus Glutinosa, Populus Nigra, Carpinus Betulus* | 3220 |

### 2.2. PlanetScope Satellite Imagery and Preprocessing

In this research, the basis for mapping allergenic tree species is commercial PlanetScope (PS) data. The imagery used in this analysis was obtained from Planet Inc., the global leader in satellite constellations, which operates over 170 orbiting satellites [23]. Many of these satellites are in a sun-synchronous orbit with 4- to 8-band radiance products at 3–4 m spatial resolution. According to Roy et al. [22], the global median average revisit interval of PlanetScope observations is one day, whereby 71.8% of land is acquired with a less than 36 h average revisit interval. In such a way, increased coverage of satellite imagery

acquired from microsatellites provides more opportunities for obtaining images with 0% cloud coverage.

PlanetScope satellites have been equipped, depending on the generation of the sensor system, since 2016 with four bands (Blue, Green, Red, Near Infrared—NIR; PS2 and PS2.SD generation), while the new generation—since 2019 (SuperDove; PSB.SD) has enabled eight-bands products (Coastal Blue, Green I, Yellow, and Red-Edge spectral bands are added) [24]. Based on the newly added spectral bands, the Red-Edge band played an important role in improving the accuracy of crop classification, due to the region where the spectral reflectance of green vegetation rises rapidly [35]. For this research, we selected six PlanetScope eight-band PSB.SD images that were acquired in a period between April and July, with 0% cloud coverage. Furthermore, the PSB.SD imagery used in this study (Table 2) included Ortho Tile Analytic (Level 3A) data products which are orthorectified, and preprocessing has been made (i.e., sensor, radiation, atmospheric, and geometric correction).

**Table 2.** The main characteristics of the PSB.SD sensor and dates used in this research.

| Parameter | Value | |
|---|---|---|
| Spectral Band | Coastal Blue (431–452 nm) Blue (465–515 nm) Green I (513–549 nm) Green (547–583 nm) | Yellow (600–620 nm) Red (650–680 nm) Red-Edge (697–713 nm) Near Infrared (845–885 nm) |
| Date | 6 April 2022 30 April 2022 15 May 2022 | 4 June 2022 18 June 2022 2 July 2022 |
| Spatial resolution | 3 m | |
| Data product | Ortho Tile Analytic Product Level 3A | |

Vegetation indices (VI) derived from RS satellite imagery are useful for evaluating vegetation cover, growth, and other characteristics [36]. Along with the four typical spectral bands (i.e., blue, green, red, and NIR), the PSB.SD sensor provides an additional four multi-spectral bands (see Section 2.2), which are narrowly focused on a particular range of the electromagnetic spectrum, which can be used for conservation, environmental monitoring, and other applications. Therefore, five different vegetation indices were used in this research, including NDVI (normalized difference vegetation index), NDWI (normalized difference water index), EVI (enhanced vegetation index), SAVI (soil adjusted vegetation index), and GNDVI (green normalized vegetation index). These indices provide information on vegetation characteristics such as chlorophyll content, cover, and leaf area [31]. The specific indices used in the research are listed in Table 3.

**Table 3.** Overview of the vegetation indices calculated from the PlanetScope (PS) imagery.

| Vegetation Index | PS Bands Used | Reference |
|---|---|---|
| NDVI | $\dfrac{B8 - B6}{B8 + B6}$ | [37] |
| NDWI | $\dfrac{B8 - B4}{B8 + B4}$ | [38] |
| EVI | $2.5 \cdot \dfrac{B8 - B6}{B8 + 6.0 \cdot B6 - 7.5 \cdot B2 + L^*}$ | [39] |
| SAVI | $\dfrac{B8 - B6}{B8 + B6 + 0.5} \cdot 1.5$ | [40] |
| GNDVI | $\dfrac{B4 - B6}{B4 + B6}$ | [41] |

* The L factor is a coefficient that accounts for variations in terrain conditions and vegetation cover. It ranges from 0, indicating dense vegetation cover, to 1, representing areas without vegetation. In this research, the L factor was set to 0.5 [42].

This research also employed the gray level co-occurrence matrix (GLCM) in order to incorporate texture information in the mapping of allergenic tree species in urban green spaces. The GLCM function, introduced by Haralick et al. [43], analyzes the texture of an image by determining the frequency of specific value-pair combinations and spatial relationships among pixels in the image. This function is useful for extracting important textural information from an image while reducing the number of correlated features [44]. Each spectral band was segmented individually, and multiple textural variables were extracted (using the GLCM measures). However, after using multiple feature selection (FS) methods (Section 3.2) with the above-mentioned GLCM features, only three GLCM features were chosen in a final classification, namely, GLCM mean, variance, and correlation [45]. These texture features were calculated (Equations (1)–(3)) as follows [46,47]:

$$\text{GLCM Mean} \; = \; \sum_{i\,=\,2}^{2N_g} i P_x \; + \; y^{(i)} \tag{1}$$

$$\text{GLCM Variance} \; = \; \sum_i \sum_j (i \; - \; \mu)^2 \, P(i,j) \tag{2}$$

$$\text{GLCM Correlation} \; = \; \sum_{i,j=0}^{N-1} P_{i,j} \left[ (i - \mu_i)(i - \mu_j) \big/ \sqrt{(\sigma_i^2)(\sigma_j^2)} \right] \tag{3}$$

where $P_{i,j}$ are the probability values occurring in adjacent within a defined window or neighborhood in the original image, and i is the column label and j is the row label of the GLCM [47]. $P_x(i)$ is equal to the i-th entry in the GLCM matrix retrieved by the row sums of $P_{i,j}$.

## 3. Methods

### 3.1. Algorithms for Image Classification

In the present research, three supervised classification algorithms (i.e., random forest (RF), extreme gradient boosting (XGB), and multi-layer perceptron (MLP)) were selected. The algorithms evaluated are those included in the 'randomForest', 'xgboost', and 'keras' packages using R statistical software, respectively.

The RF classifier works by constructing an ensemble of decision trees that each make predictions about the class label of a given pixel in an image. The decision trees' majority vote determines the pixel's final class label. It was introduced by Breiman [48] and has been found to perform well with high-dimensional data and resist overfitting, making it a robust choice for classification tasks [49,50]. There are two parameters that have to be chosen before training the RF model. The *mtry* parameter sets the number of variables used to split the nodes where each node in the decision tree is split using a random selection. Furthermore, the *ntree* parameter defines the number of trees in the forest [49].

Extreme gradient boosting (XGB) is a regularized version of traditional boosting techniques that belong to the CART (classification and regression trees) family. It is an implementation of the gradient boosting algorithm, which involves training a series of decision trees on the data and combining their predictions to make a final prediction [51]. The final classification is a combination of the improvements made by all of the previous trees. Developed by Chen and Guestrin [52], XGB is particularly useful in remote sensing because it can handle large datasets with high dimensional features and can also handle missing values in the data.

Furthermore, an ensemble of feedforward NN, multi-layer perceptron (MLP), was used for the allergenic tree species classification. The architecture of MLP includes three layers of nodes: an input layer, one or more hidden layers, and an output layer [53]. The input layer receives input data (e.g., a dataset of labeled satellite images), which is then processed by the hidden layers using weighted connections and activation functions, and finally produces output data at the output layer. The hidden layers process the input data using a set of weights and activation functions, which are used to transform the input data into a more suitable form for the output layer of the process. In a multi-layer perceptron

(MLP), the strength of connections between nodes is adjusted during training to reduce the difference between the predicted output and the actual output [54].

### 3.2. Feature Selection Methods

As mentioned in Section 2, this research used multitemporal PS imagery (Section 2.2.), as well as vegetation indices and texture features derived from satellite imagery. While additional derived input features, such as textural measures and vegetation indices, can provide valuable information for classifying Earth observation image data, they can also lead to a large number of irrelevant and redundant features [55]. In a highly dimensional dataset, variables that are correlated or redundant can introduce noise into the dataset, which can negatively impact prediction accuracy [56]. To address this issue, various feature selection methods are used to select a subset of relevant variables for the classification task [57]. Many R packages provide RF variable selection procedures (e.g., *caret* [58], *VSURF* [59], *boruta* [60], *varSelRF* [61]); hence, three feature selection (FS) methods were evaluated for mapping allergenic tree species using PS imagery: mean decrease accuracy (MDA), mean decrease Gini (MDG), and variable selection using random forests (VSURF).

Mean decrease accuracy (MDA) is an FS method that is based on the idea of evaluating how much the accuracy of a machine learning model decreases when a feature is removed from the dataset [31]. The feature that causes the least decrease in accuracy is considered the most important feature, and features that cause a large decrease in accuracy are considered to be less important. The resulting importance scores can be used to rank the features and select a subset of the most important features [62].

Similar to MDA, mean decrease Gini (MDG) is a feature selection method that uses the Gini impurity as the measure of feature importance instead of using accuracy. Gini impurity is a measure of the degree of homogeneity of the classes in a set of data, and it is commonly used in decision tree-based models [63]. The feature that causes the least decrease in Gini impurity is considered the most important feature, and features that cause a large decrease in Gini impurity are considered to be less important [64].

Variable selection using random forests (VSURF) belongs to a stepwise selection technique that uses the two-stage strategy based on backward elimination and then forward selection [65]. The method also uses the Gini impurity criterion as the measure of feature importance. It then generates a random forest model to measure the relationship between the features and the target variable and uses this information to select the most important features. The result is a subset of important features, which can be used for further analysis or modeling [59].

In this research, RF was firstly used as a classifier with the above-mentioned FS methods in order to select the most pertinent input features for mapping of allergenic tree species, i.e., six PS imagery with eight spectral bands, five vegetation indices per date, and three texture features per date and band were used as input features. The result was a subset of the most important features for each FS method, which were then used as input features using machine learning algorithms described in Section 3.1.

### 3.3. Accuracy Assessment

In order to divide the reference dataset (see Section 2.1.) into two parts: a training set (70%), which was used for fitting models, and a test set (30%) used for evaluating generalization error in the final selected model, a stratified random sampling design was used [66]. This sampling procedure was repeated ten times for more robust results. In addition, 10% of the training samples were used as validation data for the calculation of the MLP loss function [67].

Lastly, a confusion matrix was calculated for three classification scenarios (CS) based on our ground control points. Hence, accuracy was compared across the first classification scenario (CS1) which used single-date PS imagery with VI, second and third classification scenarios (CS2 and CS3) which used multitemporal PS imagery with VI, and GLCM and VI, respectively. In CS1, the PS from 15 May 2022 was used, whereas CS2 and CS3 used all available PS imagery. The results were also evaluated using traditional accuracy metrics such as overall accuracy (OA), a Kappa coefficient (Kappa), producer's accuracy (PA), and user's accuracy (UA). In addition, the F1 score and figure of merit (FoM) were also calculated.

The F1 score is a measure of the balance between PA and UA, where PA is the proportion of true positive predictions (correctly identified classes) out of all positive predictions, and UA is the proportion of true positive predictions out of all actual positive reference samples. A higher F1 score indicates a better balance between PA and UA, and a better overall performance of the model. The F1 score was calculated as follows:

$$F1 \; = \; 2 \cdot \frac{PA \cdot UA}{PA \; + \; UA} \tag{4}$$

A figure of merit (FOM) is a single value that summarizes the performance of the system or model across multiple performance metrics (i.e., overall accuracy, omission, commission), and it is often used as a way to compare the performance of different systems or models:

$$FoM \; = \; \frac{OA}{OA + OM + CM} \tag{5}$$

where OA represents overall accuracy, and OM and CM refer to the errors of omission and commission, respectively.

## 4. Results and Discussion

### 4.1. Most Suitable Classification Method

Overall, three classification scenarios (i.e., single-date PS imagery with VI—CS1, multitemporal PS imagery with VI—CS2, and multitemporal PS imagery with VI and GLCM—CS3) were computed using RF, XGB, and MLP classifiers. Table 4 shows the OA, Kappa coefficients, F1 score, and FoM for each classification scenario (CS). For each CS, ten random splits were performed to obtain ten corresponding validations (Table 4).

**Table 4.** Quantitative assessment results (i.e., overall accuracy, Kappa coefficient, F1 score, and figure of merit) for different classification scenarios (the assessment numbers in bold denote the best results among different classification scenarios).

| | Method | OA | K | Ash | | Birch | | Plane tree | | Other | |
|---|---|---|---|---|---|---|---|---|---|---|---|
| | | | | F1 | FoM | F1 | FoM | F1 | FoM | F1 | FoM |
| CS1 | RF | **52.17** | **0.36** | **0.49** | **0.34** | **0.60** | **0.39** | 0.57 | 0.38 | **0.43** | **0.32** |
| | XGB | 50.25 | 0.34 | 0.48 | 0.33 | 0.57 | 0.37 | 0.55 | 0.36 | 0.41 | 0.30 |
| | MLP | 49.36 | 0.33 | 0.37 | 0.29 | 0.53 | 0.35 | **0.58** | **0.39** | **0.43** | 0.31 |
| CS2 | RF | 69.65 | 0.59 | 0.68 | 0.52 | 0.72 | 0.55 | 0.80 | 0.63 | **0.58** | **0.45** |
| | XGB | 70.27 | **0.60** | 0.68 | 0.52 | **0.73** | **0.56** | **0.82** | **0.66** | 0.57 | **0.45** |
| | MLP | **70.29** | **0.60** | **0.70** | **0.54** | 0.71 | 0.54 | **0.82** | 0.65 | **0.58** | **0.45** |
| CS3 | RF | 71.20 | 0.62 | 0.70 | 0.54 | **0.74** | 0.57 | 0.81 | 0.66 | 0.59 | 0.46 |
| | XGB | **73.13** | **0.64** | **0.72** | **0.56** | **0.74** | **0.59** | **0.85** | **0.71** | **0.60** | **0.48** |
| | MLP | 72.07 | 0.63 | 0.71 | 0.55 | 0.73 | 0.57 | 0.84 | 0.69 | **0.60** | 0.47 |

Interestingly, as shown in Table 4, different classifiers produced the best results depending on the classification scenario (i.e., CS1 used single-date PS imagery, CS2 used multitemporal (MT) PS imagery, and CS3 used MT PS imagery with ancillary data). In such a way, RF, MLP, and XGB performed the best in CS1, CS2, and CS3, respectively. These results confirm that an RF classifier produces solid results in complex areas mostly using single-date imagery and NDVI [29,68], whereas the boosting classifier or neural network (i.e., XGB or MLP, respectively) require more training data. Similarly, Le Louarn et al. [69] achieved better results using an RF than an SVM classifier regardless of the classification scheme using a single date and bi-temporal Pleiades imagery. Furthermore, by increasing the number of training samples, MLP and RF achieved the best results for the classification of six tree species [70]. Ballanti et al. [71] recommend more than 100 training samples for every tree species in order to properly train a SVM and RF classifiers. In our research, the classification scenario with the highest number of input features (i.e., CS3) demonstrated the strength of the XGBoost classifier. Although XGB is a relatively new classifier in the field of remote sensing and its application in tree species detection has been limited, an increasing trend towards its usage has been reported in tree species classification, especially when dealing with high-dimensional data [72]. These contrasting requirements and behaviors of RF, XGB, and MLP have important implications for our study. For research with limited data, RF might be the most effective choice. Conversely, if a large volume of training data is available, more complex models such as XGB or MLP could potentially deliver superior performance, capturing subtle patterns and relationships in the data that RF might miss. The overall accuracy increased from CS1 to CS2 and CS3 using multitemporal PS imagery. In this context, since vegetation indices are influenced by many factors other than vegetation (e.g., soil moisture, atmospheric conditions, and shadows), MT PS data produced an improvement to capture changes in vegetation cover and reduced the impact of non-vegetation factors. Vegetation indices derived from the MT data are related to the biological behavior of the different species during the year, i.e., the phenological pattern associated with each species [73]. Similar to this research, MLP outperformed other ML classifiers using MT PS imagery [74], but with additional GLCM features, XGB performed the best. In the study from Georganos et al. [51], XGB was found to outperform RF and SVM tested in three different study areas, mainly in larger sample sizes. Although the F1 score is mostly used to measure a model's performance where the classes are imbalanced, both F1 and FoM were calculated to assess the ability of differentiation between different allergenic tree species. From CS1 to CS3, both measures increase according to the temporal information and ancillary data added, and in CS3, the XGB classifier obtained the highest F1 and FoM values for each class, of which plane tree has the most accurately identified positive cases. Furthermore, along with the above-mentioned accuracy measures, Stehman and Foody [75] suggest reporting UA and PA, as well as their complementary measures (i.e., commission errors or false positives, and omission errors or false negatives). Therefore, from this point on, the best results from each classification scenario (i.e., RF from CS1, MLP from CS2, and XGB from CS3) are analyzed more in detail and their visualizations are shown (Table 5, Figures 2 and 3).

The lowest UA and PA values across all three classification scenarios were for other tree classes, which highlights the difficulty distinguishing the primary chosen allergenic tree species from the rest. Most misclassification occurs with the birch plant and ash. In CS1, the birch plant had the highest UA and PA values, and in CS2 and CS3, the plane trees were detected with the lowest number of false positives and false negatives. Overall, results obtained in the classification scenario three are encouraging since most of the studies focused on allergenic pollen vegetation mapping on a large scale [10,11]. If similar research (i.e., tree species mapping in urban areas) are taken into comparison, Liu et al. [76] achieved OA for each tree species ranging from 51% to 70% using integrated airborne hyperspectral and LiDAR remote sensing data. Shojanoori et al. [77] used WorldView-2 satellite data for three types of tree species and applied maximum likelihood classification (MLC) and SVM classifier. The accuracy for MLC and SVM was 62.07% and 71.53%, respectively,

and misclassification was a common error due to the spectral similarity between tree species classes. As such, our research extensively leveraged the potential of the vegetation indices and texture features in CS2 and CS3, respectively, culminating in an overall accuracy value of 73.13%. However, we acknowledge that the aggregation of multiple tree species into a single class named 'other' could have potentially influenced our reported accuracy and presents a limitation to the precision with which we can distinguish individual allergenic tree species, especially since similar research, e.g., Kopecka et al. [78] and Degerickx et al. [79] used 15 and 17 classes in their studies, respectively. Former research achieved an OA of 90.79%, whereas the latter achieved an OA of 81%.

**Table 5.** Confusion matrix of the best classification scenario (CS1–3; bottom right corner), i.e., single-date PS imagery using RF classifier, multitemporal PS imagery using MLP, and multitemporal PS imagery with ancillary data using XGB classifier.

| Classified | Ash | Birch | Reference Plane | Other | UA [%] |
|---|---|---|---|---|---|
| Ash | 366 | 50 | 90 | 189 | 52.66 |
| Birch | 99 | 552 | 118 | 244 | 54.49 |
| Plane | 184 | 103 | 498 | 161 | 52.64 |
| Other | 163 | 137 | 101 | 372 | 48.12 |
| PA [%] | 45.07 | 65.56 | 61.71 | 38.51 | CS1 |
| Classified | Ash | Birch | Reference Plane | Other | UA [%] |
| Ash | 501 | 43 | 44 | 144 | 68.44 |
| Birch | 21 | 454 | 10 | 128 | 74.06 |
| Plane | 41 | 26 | 535 | 63 | 80.45 |
| Other | 87 | 151 | 57 | 438 | 59.75 |
| PA [%] | 77.08 | 67.36 | 82.82 | 56.66 | CS2 |
| Classified | Ash | Birch | Reference Plane | Other | UA [%] |
| Ash | 599 | 31 | 35 | 145 | 73.95 |
| Birch | 37 | 652 | 22 | 201 | 71.49 |
| Plane | 51 | 16 | 699 | 64 | 84.22 |
| Other | 125 | 143 | 51 | 556 | 63.54 |
| PA [%] | 73.77 | 77.42 | 86.62 | 57.56 | CS3 |

Beyond the scope of allergenic tree species detection, the utilization of multitemporal PS imagery opens up novel possibilities for vegetation phenology monitoring. Its high spatial and temporal resolution supports extensive, long-term ecological studies across large geographical areas [80,81]. Such phenological databases can reconcile the scale discrepancy between satellite phenology and ground observations and contribute to a more profound comprehension of forest ecosystem dynamics and future climate change modeling [82,83]. In the present study, we tapped into the potential of multitemporal PS satellite imagery in ecological research. This approach addresses a gap in the usage of remote sensing with very high spatial resolution (e.g., WorldView) and moderate spatial resolution (e.g., Landsat) [84]. In doing so, our case study concerning allergenic tree species classification significantly advances our understanding of urban vegetation specifics, thereby enhancing our comprehension of urban ecosystem services [85]. In this context, the visualization of each classification scenario with two example subsets, which show the improvement in allergenic tree species detection, is shown in Figure 2.

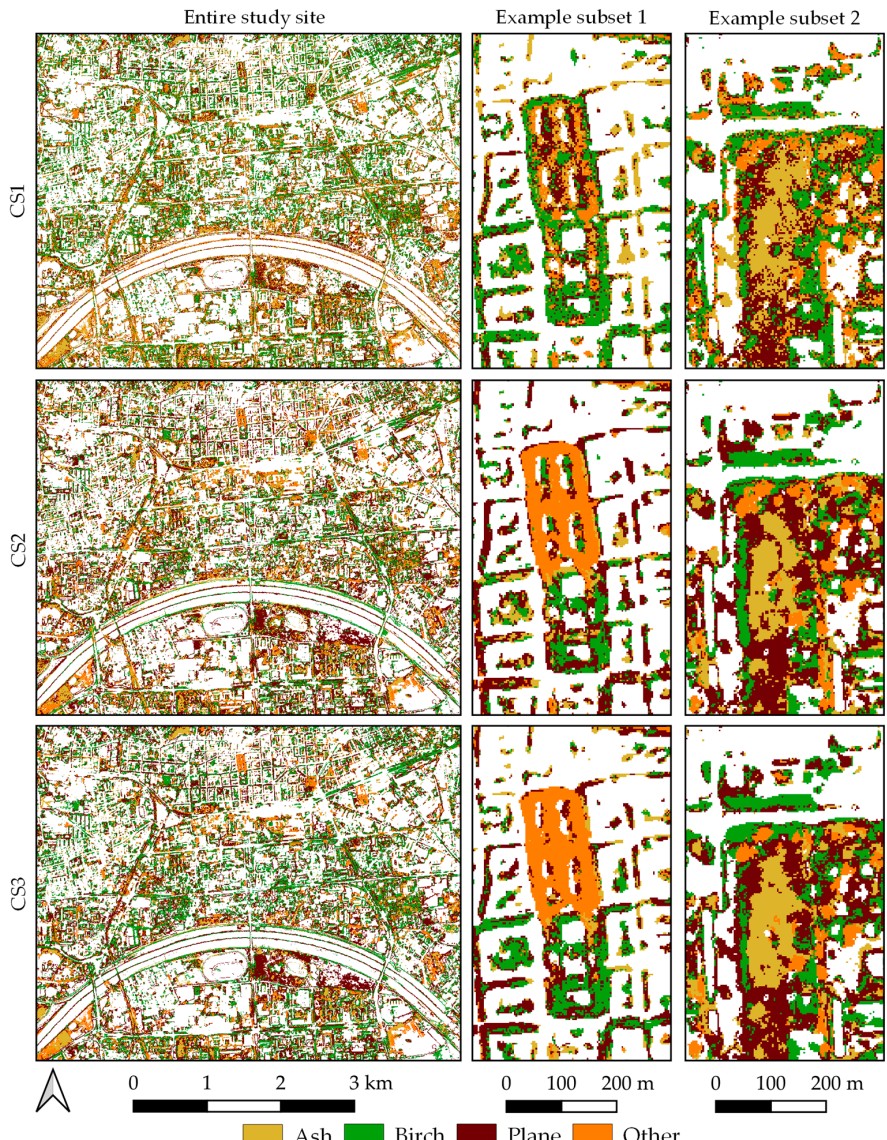

**Figure 2.** Comparison of classification maps of the entire study area (first column) generated using single-date PS imagery (CS1), multitemporal PS imagery (CS2), and multitemporal PS imagery with ancillary data (CS3), and example subset 1 (second column) and subset 2 (third column) which show improvement in allergenic tree species detection.

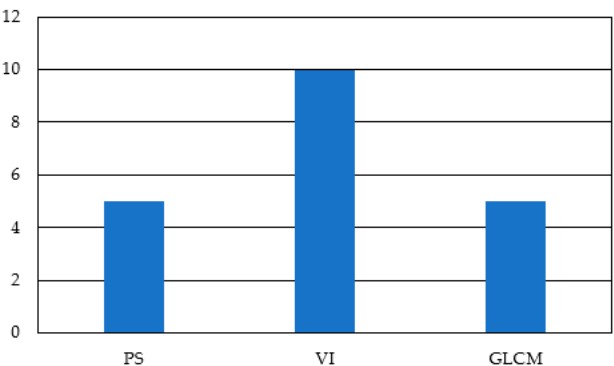

**Figure 3.** Number of input features from the VSURF feature selection method where PS indicates PlanetScope spectral band, VI vegetation indices, and GLCM texture features.

### 4.2. Most Suitable Feature Selection (FS) Method

Different FS methods were evaluated for allergenic tree species mapping using PS imagery. Table 6 shows OA and Kappa values calculated for classification scenario 3 (i.e., multitemporal PS imagery with VI and GLCM). As an outcome of the FS approach described in Section 3.2., MDA, MDG, and VSURF selected 28, 30, and 20 most important features for tree species mapping, respectively. Out of the feature selection methods evaluated, VSURF selected the smallest number of input variables. The best classification results were achieved with RF and XGB when using this VSURF subset. VSURF method has already confirmed its high performance in choosing an optimal subset of features for classification [65]. Furthermore, FS methods are not suitable for MLP classifiers, since large training datasets may avoid the overfitting problem [86,87].

**Table 6.** Accuracy results for classification scenario 3, based on different FS methods.

| | Method | MDA | | MDG | | VSURF | |
|---|---|---|---|---|---|---|---|
| | | OA | K | OA | K | OA | K |
| CS3 | RF | 69.07 | 0.59 | 68.66 | 0.58 | 69.68 | 0.60 |
| | XGB | 67.79 | 0.57 | 68.46 | 0.58 | 68.81 | 0.58 |
| | MLP | 62.92 | 0.51 | 59.68 | 0.46 | 57.82 | 0.44 |

Although the tested FS methods did not increase the classification results (Table 6) compared with the CS3 results (Section 4.1), individual input features from eight-band PS imagery can be analyzed for allergenic tree mapping. Since the VSURF method produces a list of the most pertinent input features [55], Figure 3 evaluates the number of input features grouped by the source (i.e., PS spectral band, vegetation indices, or GLCM texture feature). As expected, VI is the most represented group of input features for mapping allergenic tree species, whereas PS bands and GLCM texture features are included in a final model with five features each. In urban green areas, simple VI, which combines visible and NIR bands, has significantly improved tree detection sensitivity [36]. Out of ten VI input features, EVI and GNDVI were the most frequent. Furthermore, this research used multitemporal PS imagery acquired from the new generation of the sensor system (i.e., PSB.SD). Out of four newly added spectral bands, red-edge and coastal blue showed to be the most indicative input features. From GLCM texture features, the Variance was the most indicative input feature in a final model. As mentioned in the research by Hall–Beyer [47], variance is commonly associated with visual edges of land-cover patches. Finally, imagery acquired in April, the peak pollination month [11], proved to be the most indicative date for mapping allergenic trees.

### 4.3. Research Limitations and Prospects

The major limitations of this research include two facets, including (1) good classification results compared to the studies, which include LiDAR data combined with very high-resolution satellite imagery [76,88–90], and (2) uncertainty concerning the reference data since traditional mapping of urban green spaces (UGS) relies on field measurements, which can be time-consuming, and UGS can change quickly over time [30]. Therefore, two heatmaps were generated to identify and visualize newly discovered allergenic zones. Figure 4 clearly shows some areas not included in the reference dataset (Figure 4; first row). Later, they were detected through various classification scenarios (Figure 4; second row). The reference data were retrieved from the Web Map Service (WMS) of the Green Cadastre application, and the inventory of allergenic tree species included only the public area. A subtraction of the previously described heatmaps (i.e., ground points map minus CS3 map) resulted in a discrepancy map that shows newly detected zones where allergenic trees are located (Figure 4; third row).

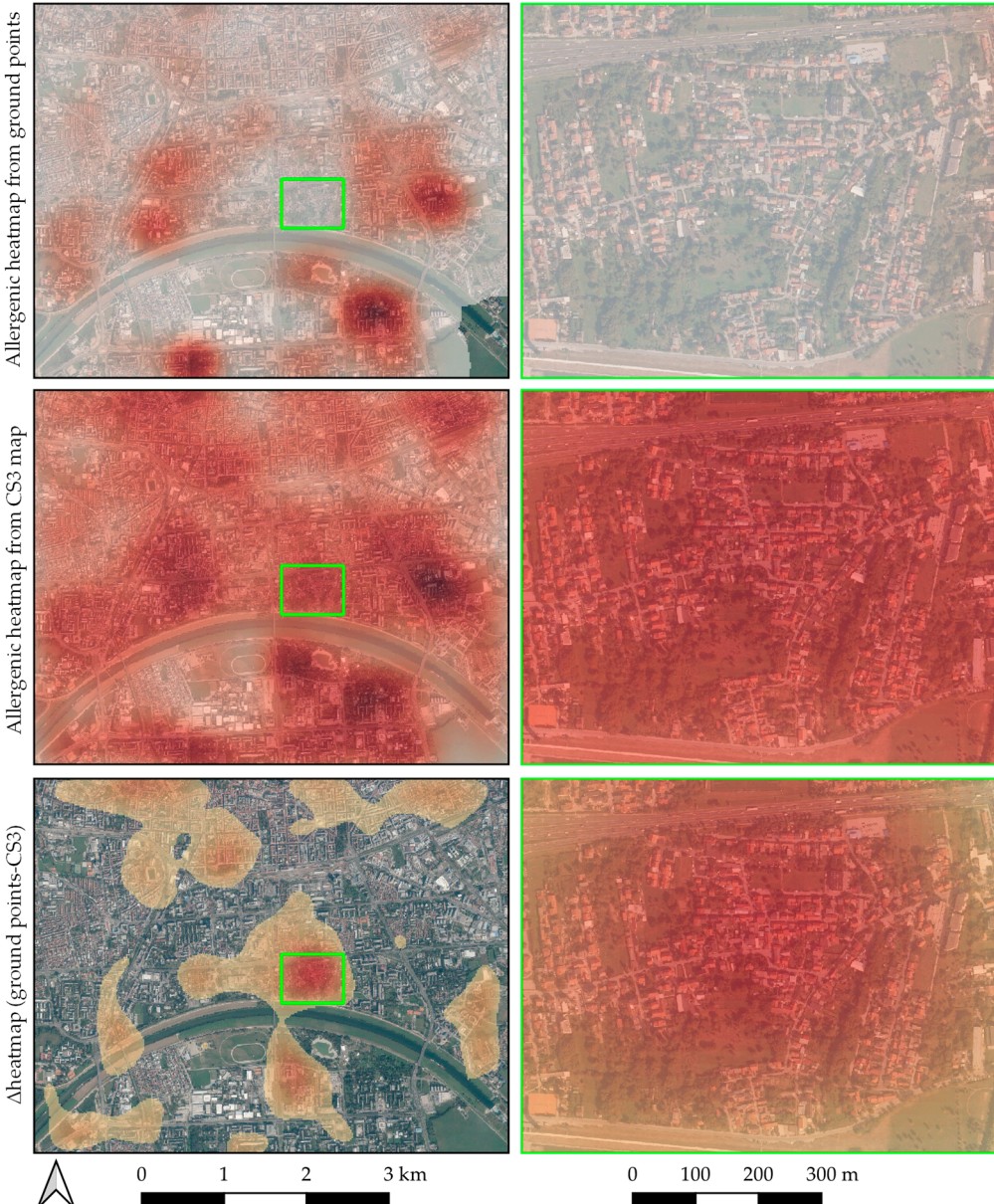

**Figure 4.** Allergenic heatmap derived from ground reference points (**first row**); classification scenario 3 (**second row**); subtraction of the ground reference points heatmap and CS3 heatmap (**third row**).

Although airborne sensors currently obtain imagery with superior spectral and spatial resolution, the unique potential of satellite sensors for conducting recurrent observations of the same area presents promising prospects for enhanced multi-temporal data acquisition and vegetation phenology-based mapping [13]. The integration of satellite imagery with additional data types, such as LiDAR or aerial photography [88,91], can further augment these capabilities. Furthermore, these datasets need to be used with deep learning (DL) methods, e.g., Hartling et al. [89] used high spatial resolution imagery in combination with LiDAR data for tree species classification in a complex urban environment, along with a DenseNet DL classifier. For the classification of eight tree species, DenseNet significantly outperformed RF and SVM regardless of training sample size.

## 5. Conclusions

Mapping allergenic tree species in highly urbanized areas is an important task as it helps for an understanding of their distribution and prevalence in urban areas. Still, most

studies concentrate on allergenic pollen vegetation mapping on a large scale, and accurate detection of individual allergenic tree species is challenging due to their smaller site and patchiness in urban green spaces. Therefore, this research tried to assess the feasibility of mapping allergenic tree species in highly urbanized areas using high-resolution PlanetScope imagery. Secondly, to evaluate and compare the RF, XGB, and MLP classifiers and feature selection methods for the accurate detection of individual allergenic tree species.

Therefore, this research showed the feasibility of using multitemporal eight-band PlanetScope imagery to detect allergenic tree species in highly urbanized areas. Furthermore, for the supervised classification of the allergenic tree species in urban areas, the XGB method, with an OA of 73.13%, outperformed the RF and MLP methods by using multitemporal PS imagery with vegetation indices and texture features. In terms of OA, classification accuracy increased by 22.88% and 2.86% from the classification on single-date imagery and multitemporal imagery with VI, respectively. In the multitemporal classification scenarios (i.e., CS2 and CS3), the plane trees were detected with the lowest number of false positives and false negatives.

The PS imagery turned out to be highly useful for tree mapping in urban green areas. By identifying and mapping allergenic tree species, city planners and public health officials can make informed decisions about tree planting and management. Future research should combine satellite imagery with very high-resolution (e.g., Pleiades, GeoEye) and LiDAR datasets with deep learning methods because it has the capability to identify and classify dominant tree species in highly complex urban environments with increased accuracy.

**Author Contributions:** Conceptualization, M.G. and I.P.; methodology, M.G. and D.D.; software, D.D.; validation, M.G. and I.P.; formal analysis, D.D.; investigation, D.D.; resources, M.G. and I.P.; data curation, M.G. and D.D.; writing—original draft preparation, D.D.; writing—review and editing, M.G., D.D. and I.P.; visualization, M.G.; supervision, M.G.; project administration, M.G.; funding acquisition, M.G. and I.P. All authors have read and agreed to the published version of the manuscript.

**Funding:** This research received no external funding.

**Data Availability Statement:** Not applicable.

**Acknowledgments:** This work was supported by the European Space Agency (ESA) as part of the scientific project: "Automatic monitoring of narrow-leaved ash (Fraxinus angustifolia Vahl) forests by remote sensing methods and Copernicus data (Grant No. RS4EST)".

**Conflicts of Interest:** The authors declare no conflict of interest.

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
