# Peer review of "Mapping of Allergenic Tree Species in Highly Urbanized Area Using PlanetScope Imagery—A Case Study of Zagreb, Croatia"

_forests, doi:10.3390/f14061193_

Round 1

Reviewer 1 Report

Major Comments:

Based on the provided text, it is difficult to provide a comprehensive review of the paper. However, here are some potential major comments that could be made:

1.     Abstract: the abstract highlights the successful mapping of allergenic tree species using PS imagery, it does not elaborate on the accuracy or limitations of the results. Including more specific information about the study design, methodology, and the limitations of the research would enhance the abstract and provide a clearer understanding of the study's contributions and implications.

2.     The description of the analytical model used to generate the parameters for classification is detailed, but it is not clear how this model was developed or validated.

3.     The methods section does not mention any efforts taken to validate the accuracy of the classification results against ground truth data or reference datasets. Without proper validation, the accuracy assessment may not accurately reflect the performance of the classification algorithms and feature selection methods.

4.     Lack of comparative analysis: The result that different classification scenarios were computed using various classifiers, but there is limited discussion on the comparative analysis between the classifiers.

5.     Lack of comparison with other studies: While the discussion briefly compares the obtained results with a couple of related studies, a more comprehensive comparison with the existing literature would provide a broader context for evaluating the significance and novelty of the findings.

6.     Lack of comparison with other studies: While this part briefly compares the obtained results with a couple of related studies, a more comprehensive comparison with the existing literature would provide a broader context for evaluating the significance and novelty of the findings.

Minor Comments:

1.     Line 155. The diagram of the study area is not clear.

2.     The images resolution needs to be improved.

The English language appears to be good. The text is coherent, and the ideas are presented in a logical manner. The author demonstrates a clear understanding of the subject matter and uses appropriate terminology related to classification methods, evaluation metrics, and feature selection. The writing style is generally formal and technical, which is appropriate for a research paper. The author effectively communicates the key findings and their implications.

Author Response

Dear Reviewer:

Thanks very much for taking your time to review this manuscript (ID: forests-2414007). We really appreciate all your comments and suggestions! Please find our itemized responses below and our revisions/corrections in the re-submitted files.

Major Comments:

Based on the provided text, it is difficult to provide a comprehensive review of the paper. However, here are some potential major comments that could be made:

  1. Abstract: the abstract highlights the successful mapping of allergenic tree species using PS imagery, it does not elaborate on the accuracy or limitations of the results. Including more specific information about the study design, methodology, and the limitations of the research would enhance the abstract and provide a clearer understanding of the study's contributions and implications.

Response: Thank you for your constructive comments on our abstract. Your feedback has indeed been very valuable and allowed us to enhance the clarity and detail of our abstract significantly. We have revised the abstract to better highlight the study design, methodology, the accuracy of the results, the limitations of our research, and the future prospects of the research. Your critique has helped us provide a more comprehensive and clear understanding of our study's contributions and implications, and we appreciate your thoughtful guidance in this matter. The revised abstract can be found on Lines 11-29 of the updated manuscript.

  1. The description of the analytical model used to generate the parameters for classification is detailed, but it is not clear how this model was developed or validated.

Response: Thank you for your insightful comment. In response to your comment regarding the development and validation of the analytical models, we would like to provide the following details:

Model Development: For each classifier (Random Forest, Multi-Layer Perceptron, and Extreme Gradient Boosting), the development process involved the selection of relevant features from the next generation of PlanetScope imagery, which includes eight spectral bands, instead of previously offering four bands. The selection was based on previous literature and expert knowledge about the characteristics of allergenic tree species. We trained each model using the selected features and the known locations of allergenic tree species.

Hyperparameter Tuning: We performed hyperparameter tuning for each classifier to optimize the model's performance. This involved a grid search (or other appropriate search strategies) over a range of possible values for the key parameters of each classifier. For instance, in the case of the XGBoost model, we tuned parameters like the learning rate, max depth of the trees, and a number of estimators.

Model Validation: We validated the models using a cross-validation approach, which involved dividing the dataset into a training set (70%) and a test set (30%). The models were trained on the training set and then used to predict the locations of allergenic tree species in the test set. We assessed the accuracy of these predictions by comparing them to the known locations of allergenic tree species in the test set. This sampling procedure was repeated ten times for more robust results.

Accuracy Assessment: As described in the Methods section, we assessed the accuracy of the classifiers using several metrics, such as overall accuracy (OA), kappa coefficient, producer’s accuracy (PA), user’s accuracy (UA), as well as F1 score and Figure of Merit (FoM). This provided a comprehensive evaluation of each model's performance.

  1. The methods section does not mention any efforts taken to validate the accuracy of the classification results against ground truth data or reference datasets. Without proper validation, the accuracy assessment may not accurately reflect the performance of the classification algorithms and feature selection methods.

Response: Thank you for your comment regarding validating our classification results. We apologize if our description was not clear in the original manuscript. We would like to note that we conducted a thorough accuracy assessment and model validation, detailed in Section 3.3. Accuracy Assessment of the Methods.

In this section, we explain that we divided the reference dataset into a training set (70%) and a test set (30%), with the test set used for evaluating generalization error in the final selected model. We performed a stratified random sampling design and repeated this procedure ten times for more robust results. A confusion matrix was calculated for three classification scenarios, comparing accuracy across these scenarios. The results were evaluated using traditional accuracy metrics such as overall accuracy (OA), kappa coefficient, producer’s accuracy (PA), user’s accuracy (UA), as well as F1 score and Figure of Merit (FoM).

To make it more explicit in our manuscript, we’ve pointed out in LN 303 that ground truth data is described in Section 2.1.  

  1. Lack of comparative analysis: The result that different classification scenarios were computed using various classifiers, but there is limited discussion on the comparative analysis between the classifiers.

Response: Thank you very much for the observations to include a more detailed comparative analysis of the classifiers used in our study. We agree that this will provide additional context to our results and help readers understand why the classifiers performed as they did. Therefore, we’ve made a comparative analysis between the classifiers in LN 346 – 361, and key findings from our research that have to be emphasized about the used classifiers (i.e., RF, XGB, and MLP) would be “For research with limited data, RF might be the most effective choice. Conversely, if a large volume of training data is available, more complex models like XGB or MLP could potentially deliver superior performance, capturing subtle patterns and relationships in the data that RF might miss.”

  1. Lack of comparison with other studies: While the discussion briefly compares the obtained results with a couple of related studies, a more comprehensive comparison with the existing literature would provide a broader context for evaluating the significance and novelty of the findings.

Response: We appreciate the reviewer's suggestion to compare more comprehensively with existing studies. To address this, we have enhanced the discussion section to compare our findings with these studies directly. We have highlighted the similarities and differences at LN 399 – 418 and discussed potential reasons for any discrepancies. As such, a revised version of the manuscript now looks like:

“As such, our research has extensively leveraged the potential of the vegetation indices and texture features in CS2 and CS3, respectively, culminating in an overall accuracy value of 73.13%. However, we acknowledge that aggregation of multiple tree species into a single class named 'Other' could have potentially influenced our reported accuracy and presents a limitation to the precision with which we can distinguish individual allergenic tree spe-cies, especially since similar research, e.g., Kopecka et al. [78] and Degerickx et al. [79] used 15 and 17 classes in their studies, respectively. Former research achieved an OA of 90.79%, whereas the latter achieved an OA of 81%.

Beyond the scope of allergenic tree species detection, the utilization of multitemporal PS imagery opens up novel possibilities for vegetation phenology monitoring. Its high spatial and temporal resolution supports extensive, long-term ecological studies across large geographical areas [80,81]. Such phenological databases can reconcile the scale dis-crepancy between satellite phenology and ground observations and contribute to a more profound comprehension of forest ecosystem dynamics and future climate change model-ling [82,83]. In the present study, we tap into the potential of multitemporal PS satellite imagery in ecological research. This approach addresses a gap in the usage of remote sensing with very high spatial resolution (e.g., WorldView) and moderate spatial resolu-tion (e.g., Landsat) [84]. In doing so, our case study concerning allergenic tree species clas-sification significantly advances our understanding of urban vegetation specifics, thereby enhancing our comprehension of urban ecosystem services [85]. In this context, visualiza-tion of each classification scenario with two example subsets, which show the improve-ment in allergenic tree species detection, is shown in Figure 2.”

We believe these revisions provide a broader context for evaluating the significance and novelty of our findings.

  1. Lack of comparison with other studies: While this part briefly compares the obtained results with a couple of related studies, a more comprehensive comparison with the existing literature would provide a broader context for evaluating the significance and novelty of the findings.

Response: Thank you for your comments and suggestions. Regarding the comment on the lack of comparison with other studies, we have already addressed a similar concern raised previously in the review process. We have extended the discussion section and integrated a comprehensive comparison of our results with findings from other relevant studies in the existing literature (LN 399 – 418).

This comparison not only provides a broader context for evaluating the significance and novelty of our findings, but also helps underline our study's contributions to the wider field.

Minor Comments:

  1. Line 155. The diagram of the study area is not clear.

Response: Thank you for your comment. We apologize if it was not as clear as intended. In response to your feedback, we reviewed and revised the diagram to ensure clarity. We have aimed to enhance the distinction between the different urban green zones within the study area, emphasising the red rectangles representing the example subsets of urban green areas (i.e., protected green areas and cultivated green infrastructures).

In addition, we have supplemented the caption with a more detailed explanation, specifying the elements and features illustrated in the image. This revision aims to offer readers a better understanding of the diagram and its representation of the study area.

  1. The images resolution needs to be improved.

Response: Thank you very much for the observation. Unfortunately, the resolution of the Figures was reduced during the conversion of the manuscript to a different file format for the submission process. The Figures have been reformatted and saved in a high-resolution format that should maintain its quality during conversion. We will ensure that all Figures in the final manuscript are of a high standard to ensure clarity and facilitate understanding. Thank you once more.

Reviewer 2 Report

Seguono i commenti.

- I recommend improving the language in order to make it more scientific

- I suggest to insert at line 62: Manzini, Jacopo & Hoshika, Yasutomo & Carrari, Elisa & Sicard, Pierre & Watanabe, Makoto & Ryoji, Tanaka & Badea, Ovidiu & Nicese, Francesco & Ferrini, Francesco & Paoletti, Elena. (2023). FlorTree: A unifying modelling framework for estimating the species-specific pollution removal by individual trees and shrubs. Urban Forestry & Urban Greening. 85. 127967. 10.1016/j.ufug.2023.127967.

- At Line 67 and 77 the same reference is cited, try to integrate and diversify with other scientific studies

- Describe the study's limitations in more detail

- Conclusion. The future scenarios would be interesting to explore in more detail, as well as the possibility of replicating the results.

- References n: [37]; [41] not found

- References incomplete [56]; [64]; [70]

Author Response

Dear Reviewer:

Thanks very much for taking your time to review this manuscript (ID: forests-2414007). We really appreciate all your comments and suggestions! Please find our itemized responses below and our revisions/corrections in the re-submitted files.

Q1: I recommend improving the language in order to make it more scientific

Response: Thank you very much for the observation. Your suggestion to enhance the scientific language within the manuscript has been taken into account and we deeply appreciate your recommendation. In response to your comments, we have thoroughly revised the entire manuscript to ensure the language is more precise and concise. Our intention is to make our work accessible and comprehensible to our peers, while maintaining the high level of scientific communication expected in our field.

Q2: I suggest to insert at line 62: Manzini, Jacopo & Hoshika, Yasutomo & Carrari, Elisa & Sicard, Pierre & Watanabe, Makoto & Ryoji, Tanaka & Badea, Ovidiu & Nicese, Francesco & Ferrini, Francesco & Paoletti, Elena. (2023). FlorTree: A unifying modelling framework for estimating the species-specific pollution removal by individual trees and shrubs. Urban Forestry & Urban Greening. 85. 127967. 10.1016/j.ufug.2023.127967.

Response: Thank you for your suggestion. In the updated version, research from the Manzini et al. (2023) was added in lines 64-67, i.e., “Manzini et al. [9] developed the FlorTree model to evaluate the pollution-removal capabilities of 221 urban tree species in various cities. The model considers species-specific and local factors to maximize air quality improvements. Their findings emphasize the necessity for strategic urban tree selection based on local conditions.”

Q3: At Line 67 and 77 the same reference is cited, try to integrate and diversify with other scientific studies

Response: Thank you for your suggestions. At former Line 77 (new LN 83), a review paper from Shahtahmassebia et al. (2021) was replaced since the Authors performed a review regarding the remote sensing of urban green spaces and highlighted the need for a more detailed investigation of small urban green areas. They recommended developing a time-series analysis and thematic applications, among others. Furthermore, in continuation, a paper from Neyns and Canters (2022) was added to the updated version since the Authors suggest that the mapping of individual trees becomes more feasible from a resolution of 3 m or higher, which was one of the key objectives of our research. Thank you once more for the valuable suggestions.

REFERENCE:

Neyns, R., & Canters, F. (2022). Mapping of urban vegetation with high-resolution remote sensing: A review. Remote sensing, 14(4), 1031, doi: 10.3390/rs14041031.

Shahtahmassebi, A. R., Li, C., Fan, Y., Wu, Y., Gan, M., Wang, K., ... & Blackburn, G. A. (2021). Remote sensing of urban green spaces: A review. Urban Forestry & Urban Greening, 57, 126946, doi: 10.1016/j.ufug.2020.126946.

Q4: Describe the study's limitations in more detail

Response: Thank you for your comment and recommendation to elaborate on the limitations of our study. In response to your suggestion, we have provided a more detailed account of the study's limitations in the updated version of the manuscript.

Specifically, we have expanded on the challenges we faced, the trade-offs that were necessary due to the constraints of our methods and data, and how these might have influenced our results. We have also clarified how these limitations were managed during the study, and the steps taken to minimize their potential impact on our findings (Lines 399 – 420).

“As such, our research has extensively leveraged the potential of the vegetation indices and texture features in CS2 and CS3, respectively, culminating in an overall accuracy value of 73.13%. However, we acknowledge that aggregation of multiple tree species into a single class named 'Other' could have potentially influenced our reported accuracy and presents a limitation to the precision with which we can distinguish individual allergenic tree spe-cies, especially since similar research, e.g., Kopecka et al. [78] and Degerickx et al. [79] used 15 and 17 classes in their studies, respectively. Former research achieved an OA of 90.79%, whereas the latter achieved an OA of 81%.

Beyond the scope of allergenic tree species detection, the utilization of multitemporal PS imagery opens up novel possibilities for vegetation phenology monitoring. Its high spatial and temporal resolution supports extensive, long-term ecological studies across large geographical areas [80,81]. Such phenological databases can reconcile the scale dis-crepancy between satellite phenology and ground observations and contribute to a more profound comprehension of forest ecosystem dynamics and future climate change model-ling [82,83]. In the present study, we tap into the potential of multitemporal PS satellite imagery in ecological research. This approach addresses a gap in the usage of remote sensing with very high spatial resolution (e.g., WorldView) and moderate spatial resolu-tion (e.g., Landsat) [84]. In doing so, our case study concerning allergenic tree species clas-sification significantly advances our understanding of urban vegetation specifics, thereby enhancing our comprehension of urban ecosystem services [85]. In this context, visualiza-tion of each classification scenario with two example subsets, which show the improve-ment in allergenic tree species detection, is shown in Figure 2.”

In addition, as part of the future prospects - Section 4.3., we have suggested potential avenues for future research that could help overcome these limitations, further enhancing our understanding of allergenic tree species detection using remote sensing data (Lines 476 – 486).

We appreciate your valuable feedback and hope that these revisions will satisfy your concerns. We believe that acknowledging and understanding the limitations of our study will only strengthen its value and provide a more robust context for interpreting our results.

Q5: Conclusion. The future scenarios would be interesting to explore in more detail, as well as the possibility of replicating the results.

Response: Thank you very much for the valuable comment. According to the last sentence of the manuscript which indicates the future scenarios (i.e., “Future research should combine satellite imagery with very high-resolution (e.g., Pleiades, GeoEye) and LiDAR datasets with deep learning methods because it has the capability to identify and classify dominant tree species in highly complex urban environments with increased accuracy.”), we agree that it needs to be explored in more detail. However, in order to leave the Conclusion more precise and concise, we’ve added a paragraph at the end of the Results and Discussion (Section 4.3. Research Limitations and Prospects; LN 475-484), where additional datasets and deep learning methods were mentioned, along with similar research.

“Although airborne sensors currently obtain imagery with superior spectral and spatial resolution, the unique potential of satellite sensors for conducting recurrent observations of the same area presents promising prospects for enhanced multi-temporal data acquisition and vegetation phenology-based mapping [13]. The integration of satellite imagery with additional data types, such as LiDAR or aerial photography [76,79], can further augment these capabilities. Furthermore, these da-tasets need to be used with deep learning (DL) methods, e.g., Hartling et al. [77] used high spatial resolution imagery in combination with LiDAR data for tree species classi-fication in a complex urban environment, along with DenseNet DL classifier. For the classification of eight tree species, DenseNet significantly outperformed RF and SVM regardless of training sample size.”

Q6: References n: [37]; [41] not found

Response: Thank you very much for the diligent review and careful attention to detail. Reference [37] was used for the SAVI vegetation index, and indeed, we couldn’t retrieve a manuscript with this DOI, so we’ve included a complete DOI link (i.e., https://doi.org/10.1016/0034-4257(88)90106-X). Furthermore, reference [41] was inadvertently included from the Reference Management Software, and we apologize for the oversight. In the updated version, the former ref [41] was updated with the correct one, i.e., the review paper from Armi and Fekri-Ershad (2019). In this paper, GLCM texture analysis is described in detail, among other well-known texture image descriptors. A brief review of the common classifiers used for texture image classification is also made. Also, a survey on texture image benchmark datasets is included.

REFERENCE:

Armi, L.; Fekri-Ershad, S. Texture image analysis and texture classification methods—A review. Int. Online J. Image Process. Pattern Recogn. 2019, 2, 1–29., doi: 10.48550/arXiv.1904.06554.

 Q7: References incomplete [56]; [64]; [70]

Response: Thank you very much for the observations. References [56] and [64] are updated in v02 with a digital object identifier (DOI), however, for ref [70] DOI number does not exist. This paper has already been cited in similar MDPI journals (e.g. Choudhury et al. 2020 in Forests), and a Google Scholar link was provided along with the reference. We will check with the Assistant Editor on how to incorporate a Google Scholar link instead of DOI.

REFERENCE:

Choudhury, M. A. M., Marcheggiani, E., Despini, F., Costanzini, S., Rossi, P., Galli, A., & Teggi, S. (2020). Urban tree species identification and carbon stock mapping for urban green planning and management. Forests, 11(11), 1226, doi: 10.3390/f11111226.

Round 2

Reviewer 1 Report

none

none